# Quantitative Evaluation of Caries and Calculus with Ultrahigh-Resolution Optical Coherence Tomography

**DOI:** 10.3390/bioengineering10111317

**Published:** 2023-11-15

**Authors:** Tai-Ang Wang, Nguyễn Hoàng Trung, Hsiang-Chieh Lee, Cheng-Kuang Lee, Meng-Tsan Tsai, Yen-Li Wang

**Affiliations:** 1Graduate Institute of Photonics and Optoelectronics, National Taiwan University, Taipei 10617, Taiwan; d10941014@ntu.edu.tw (T.-A.W.); hclee2@ntu.edu.tw (H.-C.L.); 2Department of Electrical Engineering, Chang Gung University, Taoyuan 33302, Taiwan; trung12ly@gmail.com; 3Department of Electrical Engineering, National Taiwan University, Taipei 10617, Taiwan; 4NVIDIA AI Technology Center, NVIDIA, Taipei 11492, Taiwan; ckl@nvidia.com; 5Department of Dermatology, Chang Gung Memorial Hospital, Linkou 33305, Taiwan; 6Department of Periodontics, Chang Gung Memorial Hospital, Taoyuan 33378, Taiwan; 7College of Medicine, Chang Gung University, Taoyuan 33302, Taiwan

**Keywords:** caries, calculus, optical coherence tomography, scattering, roughness

## Abstract

Dental caries on the crown’s surface is caused by the interaction of bacteria and carbohydrates, which then gradually alter the tooth’s structure. In addition, calculus is the root of periodontal disease. Optical coherence tomography (OCT) has been considered to be a promising tool for identifying dental caries; however, diagnosing dental caries in the early stage still remains challenging. In this study, we proposed an ultrahigh-resolution OCT (UHR-OCT) system with axial and transverse resolutions of 2.6 and 1.8 μm for differentiating the early-stage dental caries and calculus. The same teeth were also scanned by a conventional spectral-domain OCT (SD-OCT) system with an axial resolution of 7 μm. The results indicated that early-stage carious structures such as small cavities can be observed using UHR-OCT; however, the SD-OCT system with a lower resolution had difficulty identifying it. Moreover, the estimated surface roughness and the scattering coefficient of enamel were proposed for quantitatively differentiating the different stages of caries. Furthermore, the thickness of the calculus can be estimated from the UHR-OCT results. The results have demonstrated that UHR-OCT can detect caries and calculus in their early stages, showing that the proposed method for the quantitative evaluation of caries and calculus is potentially promising.

## 1. Introduction

Oral health is vital and impacts general health and quality of life. Oral diseases are amongst the most neglected diseases. They do not just affect the teeth but can also be related to systemic diseases [1,2,3]. Dental caries and periodontitis are the two most prevalent oral diseases. Dental caries is generated by the interaction of bacteria and carbohydrates on the surface of the crown, causing acid erosion, which gradually breaks down the hard tissue of teeth. Extensive cavities could lead to tooth loss and even possibly cause infection and sepsis, resulting in serious systemic disorders [4]. Periodontal disease is a chronic inflammatory disease caused by bacterial infection. After dental plaques mineralize into dental calculus, bacteria can attach more easily to its rough surface texture. Dental calculus is one of the main culprits harming periodontal health. Failure to properly clean the dental calculus and plaques can result in a vicious cycle of damaging the supporting tissues around the teeth, eventually leading to tooth loss [5,6]. Periodontal disease is also associated with several systemic inflammations, such as diabetes and atherosclerosis [7]. Early detection is becoming an increasingly important issue in dentistry. With early-stage detection and proper treatment of oral diseases, the lesions can be controlled effectively to avoid irreversible damage or further deterioration and expansion of the disease [8,9]. The traditional diagnostic method is mainly based on visual assessments with invasive clinical examinations such as tooth drilling and periodontal probing [10,11]. This detection relies heavily on the professional skills and experience of doctors, and has limited reliability and accuracy, with the possibility of missing marginal areas or small affected areas, which are visually obscure [12]. To assist in diagnosis, many technologies have been applied in dentistry. Radiography is considered the gold standard auxiliary diagnosis imaging tool routinely used in oral practice [13]. However, its two-dimensional images and limited resolution make it difficult to judge the subtle structural changes in the complicated oral cavity. Dental computed tomography (dental CT) can provide three-dimensional images with its ability to scan various angles, yet the lack of resolution and the concerns about radiation still limit its usage [14]. Laser-induced fluorescence is proposed for inspecting the condition of intraoral disease [15]. Nevertheless, this technique is too sensitive to ambient light, which can limit its image quality and consistency.

Optical coherence tomography (OCT) is an optical imaging technology based on a low-coherence interference that acquires volumetric images in label-free, noncontact, and nonradiative situations. Since it was first developed in 1991, after 30 years of efforts by research and development groups, OCT has been widely used in different medical research and clinical trials [16,17,18,19]. In addition, advanced functional systems and combinations with other techniques, such as polarization-sensitive OCT (PS-OCT) [20,21,22,23] and fluorescence-guided OCT [24], enable OCT to integrate structural information with other specific optical features, expanding its application in the biomedical field. Moreover, as a major development, ultrahigh-resolution OCT (UHR-OCT) has emerged as an optimal solution for OCT’s spatial resolution [25,26,27,28]. With the advances in light source technologies, short-wavelength and broadband light sources, such as femtosecond lasers, multiple super luminescent diodes, and supercontinuum lasers are utilized to optimize the axial resolution of UHR-OCT. The general axial resolution of UHR-OCT is less than 2 μm in tissue, which is much higher than that of the conventional OCT systems. Combined with the high transverse resolution obtained by applying high numerical aperture (NA) focusing in the system, UHR-OCT can be used to obtain submicron or cellular-level volumetric and sectioned structural histological images. 

The application of OCT to dentistry has become increasingly popular since 1998 [29,30,31]. Owing to the high sensitivity of OCT, its image quality and consistency are unaffected by the factors of the external environment or human judgment. Its better resolution also makes it more suitable for early diagnosis and the detection of minute initial lesions compared with traditional diagnostic methods. With its capability to perform cross-sectional imaging of microstructures, OCT can be used to observe soft- and hard-tissue diseases in the oral cavity, such as dental caries, tooth cracks, damage to the periodontal tissue, and oral cancer [32]. Multiple groups have demonstrated their results in early dental assessments using OCT [33,34,35]. The identification of early dental caries and calculus is of great significance in clinical diagnosis [36]. 

In this study, we developed a UHR-OCT system with axial and transverse resolutions of 2.6 and 1.8 μm, respectively. The UHR-OCT was utilized to image sound teeth, cavities, and calculus in the very early stages, and to observe the morphological and optical changes in the enamel. To further verify the results, quantitative evaluations were performed including the roughness and the scattering coefficient. Finally, the results of UHR-OCT were compared with the results obtained from a conventional spectral-domain OCT (SD-OCT) system with an axial resolution of 7 μm to demonstrate the potential of UHR-OCT in dentistry.

## 2. Methods

### 2.1. Setup of the UHR-OCT System

The experimental setup illustrated in Figure 1 was a homemade UHR-SD-OCT system based on a Michelson interferometer and a spectrometer devised in house. A supercontinuum laser (SuperK EXTREME, NKT Photonics, Demark) was used as the light source, ranging from the visible to near-infrared regions. The light from the light source was then collimated and passed through a variable neutral density filter (NDC-50C-4-B, Thorlabs, Newton, NJ, USA) to dynamically adjust the input optical power to maintain the balance between optimized performance and the maximum permissible exposure (MPE) limit of different samples. The integration of this filter was instrumental in effectively managing the power of the incident light, thereby preventing potential damage to the sample or saturation of the intensity. A broadband beam splitter (BNPB-20B-45R-800, Lambda Research Optics, Costa Mesa, CA, USA) with a 50:50 splitting ratio was used, dividing the incident light into the sample and reference arms, respectively. The sample arm was composed of a two-axis galvanometer (GVS002, Thorlabs, Newton, NJ, USA) and an objective lens with a long working distance (M Plan Apo NIR 20X, Mitutoyo, Kawasaki, Kanagawa, Japan). In the sample arm, the diameter of the collimated beam was accurately adjusted to match the size of the mirrors of the two-axis galvanometer, thereby optimizing the lateral resolution. In the reference arm, an iris diaphragm was used to control the return power. The recombinational interference light was coupled by an achromatic lens into a single-mode fiber (SMF) which then translated the light from the interferometer to the spectrometer. The spectrometer was composed of a transmission diffraction grating (T-1200-850S, Finisar, Coherent, Saxonburg, PA, USA), an achromatic lens group, and a CMOS USB 3.0 line-scan camera as the detector (OCTOPLUS UB, e2v, Teledyne, Essex, UK). Optical simulation of the spectrometer’s design was performed with the optical design software (OpticStudio, Zemax, Ansys, Canonsburg, PA, USA). The light emitted from the SMF was dispersed using a transmission grating of 1200 lines/mm. Subsequently, the achromatic lenses were implemented for focusing the dispersive light to the camera. The pixel number of the camera was 2048 pixels with a tall-pixel size of 10 μm × 200 μm. A DAQ card (PCIe-6259, National Instruments, Austin, TX, USA) was utilized to control the galvanometers and to generate a trigger signal for grabbing frames. Here, a trigger generator was used to produce a pulsed signal, ensuring synchronization between the line-scan camera of the spectrometer and the galvanometer. In addition, a conventional-resolution SD-OCT system with an axial resolution of 7 μm was implemented for comparison. The setup of the conventional-resolution SD-OCT was described in a previous study [37]. Table 1 shows a comparison of the specifications of the two systems. The sensitivities of the conventional-resolution SD-OCT and UHR-OCT systems were 96 and 95 dB, respectively. The maximum power incident on the sample for both systems was set to ~1 mW.

### 2.2. Sample Collection and Experimental Procedure

In this study, human teeth were extracted and confirmed by experienced dental practitioners. The extracted teeth were mainly divided into three categories: sound teeth, teeth with early decay, and teeth with early calculus. There were at least 11 samples for each type (*n* ≥ 11) to ensure the accuracy of the experiment. These teeth were extracted from patients who visited Chang Gung Memorial Hospital. After extraction, the teeth were stored in water at 4 °C and then dried in the air before OCT scanning. To facilitate the OCT experimental process, the teeth were resin-bonded and fixed in a 3D-printed holder mounted on a three-axis stage. The study was conducted in accordance with the relevant regulations and guidelines approved by the Institutional Review Board of Chang Gung Medical Foundation (No. 202000421B0). All the experimental procedures were approved by the Gung Medical Foundation Institutional Review Board. In this study, the tooth samples were initially categorized into three groups: normal, early caries, and early calculus, based on the results of optical microscopy (OM) analyzed by experienced practitioners. To further corroborate the early-stage caries, some teeth underwent destructive SEM examination after OCT imaging.

### 2.3. Data Analysis

This study leveraged different methods to quantitively analyze the differences between the teeth with caries in different initial stages and sound teeth. Moreover, the structural segmentation of the early calculus on teeth was investigated.

#### 2.3.1. Surface Roughness of Dental Caries

As the first estimation, quantifying the surface roughness can be effective for observing whether the enamel surface has been affected by demineralization and remineralization [38]. Figure 2 shows the flow chart of the assessment of surface roughness. First, the background noise of the OCT images is removed, and then, the surface is scanned to evaluate the roughness coefficient [33,39]. The root mean square of the surface roughness *R_q_* can be expressed as
(1)$$R_{q}=\sqrt{\frac{1}{n}\sum_{i=0}^{n-1}{\left(z_{i}\right)}^{2}}$$
where *n* is the total number of A-scans in one B-scan, and *z_i_* is the location of the decayed area of each A-scan in terms of depth from the surface. In a previous report, the surface roughness was estimated for each B-scan according to Equation (1).

However, in our case, it may not be accurate to directly estimate the surface roughness based on Equation (1), as the results can be severely affected by the complex geometry of the tooth’s surface. To solve this issue, alternating symmetrical filters were chosen as the filtration method. They were used to filter the original form of the tooth’s surface prior to the evaluation of the surface roughness. An alternating symmetrical filter typically comprises multiple iterations of erosion and dilation operations. The process begins with an erosion operation, which eliminates minor details and noise from the image. This is subsequently followed by a dilation operation, expanding the remaining features. The process is repeated, alternating between erosion and dilation. Through the iterative application of these operations, the filter can effectively eliminate the asymmetric components while preserving the symmetric structures. This method may prove beneficial for applications such as edge detection, noise reduction, and shape analysis [40,41]. In this study, an opening filter and closing filter were used, and both had a combination of erosion and dilation operations to form the alternating symmetrical filter. The filtration method was applied to each B-scan independently. After subtracting the waviness profile of the tooth, the remaining data could be calculated to obtain the actual surface roughness.

#### 2.3.2. Scattering Coefficient of Dental Caries

The scattering characteristics could be an indicator for determining tooth decay, because the scattering effect of the enamel changes significantly upon the occurrence of the demineralization process. To verify that the UHR-OCT system could differentiate the sound teeth and early dental caries in different stages, depth-resolved pre-processing and a single scattering model combining dynamic focusing were introduced to estimate the scattering coefficient, *μ_s_* [33,42]. The depth-resolved numerical method was applied to compensate for the signal decay caused by the depth profile to improve the accuracy of evaluating the scattering coefficient. The OCT’s intensity signal based on a single scattering model with dynamic focusing *I*(*z*) can be expressed as
(2)$$I(z)\propto \sqrt{\frac{e^{-2\mu_{s}z}}{\left[1+{\left(\frac{z-z_{f}}{z_{R}}\right)}^{2}\right]}}$$

According to the Beer–Lambert law, the depth-dependent OCT signal of enamel, *I*(*z*), would be attenuated on an exponential scale [43]. The point spread function was integrated to simulate the dynamic focusing of OCT signals. In Equation (2), *z* is the depth range, while *z_f_* and *z_R_* are the depth of the location of the focal plane and the Rayleigh length, respectively.

#### 2.3.3. Segmentation of Dental Calculus

To acquire the component of tooth calculus from OCT images, the surface profile of each tooth was obtained according to the method illustrated in Figure 2. As the calculus is located on the tooth’s surface, the region of calculus can be defined as the area between the calculus and the tooth’s surfaces. Therefore, the locations of the calculus and the tooth’s surfaces for each A-scan were determined, and the distance between the two locations then corresponded to the thickness of the calculus. After estimation from a 3D dataset, the distribution of the thickness of calculus was acquired.

## 3. Results

### 3.1. System Characterization

The resolution of OCT is contingent upon the central wavelength and bandwidth. Attaining an ultrahigh axial resolution in OCT is challenging, as it necessitates complex ultra-broadband light sources. Moreover, the central wavelength of the light source exerts a significant influence. Indeed, the impact of the central wavelength on the axial resolution is even more pronounced than that of bandwidth. Thus, utilizing a central wavelength close to 800 nm with sufficient bandwidth is the most effective way to balance the axial resolution and the imaging depth. Owing to the ultra-broadband characteristics of the supercontinuum laser, the resolution of the newly developed UHR-OCT was mainly determined by the alignment of the fibers’ coupling and the design of the spectrometer. In this study, the spectrometer was designed to cover a spectral range of 675–875 nm at a central wavelength of 775 nm. Figure 3a depicts the interference spectrum received by the spectrometer. The black line represents the received spectrum of the reference arm when the sample arm was blocked, and the red line shows the original interferogram using a silver mirror as the sample. Figure 3b shows the interference signal obtained after the processes of removing the direct current (DC), resampling, and window shaping. To further reduce the dispersion effect, a software tool for compensation for dispersion was implemented in this study [44]. Subsequently, the interference data were then transformed by a fast Fourier transform to retrieve the axial point spread function (PSF), as plotted in Figure 3c. The measured axial resolution of the UHR-OCT was approximately 2.6 μm in the air, corresponding to 1.63 μm in tooth enamel tissue with an average refractive index of 1.6 [45]. The lateral resolution was measured using the United States Air Force (USAF) resolution target as the sample, as shown in Figure 3d, and the central area of the resolution target, indicated by the red square, was magnified (Figure 3e). In Figure 3e, the line pairs of elements 1 and 2 in group 8 were resolved, corresponding to 1.95 μm and 1.74 μm as the lateral resolution of UHR-OCT. Therefore, the lateral resolution of the newly developed UHR-OCT was ~1.8 μm. Although the maximum *en-face* field of view (FOV) of the UHR-OCT system could achieve 2 mm × 2 mm, the FOV was adjusted to 1 mm × 1 mm to maintain the image’s quality. For 3D imaging, 1000 B-scans, each composed of 1000 A-scans, were captured at a line rate of 50 kHz, corresponding to a frame rate of 50 Hz.

Although OCT has been widely used for diagnosing dental caries in previous reports, the resolution of OCT is a key issue in the detection of early-stage caries. To investigate the capacity of OCT for early diagnosis of caries, two OCT systems with different resolutions were implemented to scan teeth, including normal teeth and teeth with caries, and calculus for comparison. The detailed specifications are shown in Table 1. Figure 4a–c represent the OCT images of a normal tooth obtained by a conventional-resolution SD-OCT system including the *en-face* image at the tooth surface, the cross-sectional image, and the *en-face* image at the depth indicated by the white line in Figure 4b. The yellow arrows in Figure 4b,c indicate the boundary between the enamel and the dentin. In contrast, Figure 4d–g show the OCT images obtained by UHR-OCT of the same normal tooth including the *en-face* image at the tooth’s surface, the cross-sectional images at locations I and II, and the *en-face* image at the depth indicated by the white line in Figure 4f. It was difficult to scan the same region with both OCT systems, but the scanned regions were to be selected as close as possible. The red arrows in Figure 4a–d indicate the small area of damage on the tooth surface. The tooth with caries was then scanned with both OCT systems, and Figure 4h–j and Figure 4k–n represent the results obtained using the conventional-resolution OCT and UHR-OCT systems, respectively. Figure 4h,k show the *en-face* images of the tooth surface, and Figure 4i,l,m show the cross-sectional images. Figure 4j,n are the *en-face* images at the depth indicated by the white lines in Figure 4i,m. In the normal tooth results, the tooth’s surface can be clearly identified by the presence of smooth sharpness. However, as indicated by the red arrow in Figure 4i, the tooth’s surface became rough in comparison with that of Figure 4b. Furthermore, with UHR-OCT, the structure of caries could be further visualized as indicated by the red arrows in Figure 4l,m. In addition, Figure 4n shows that UHR-OCT can detect tinier changes in the tooth’s structure better than the conventional-resolution OCT system, as shown in Figure 4j.

Figure 5 shows the results of another case of caries obtained by using the conventional-resolution OCT and UHR-OCT systems. Figure 5a is a 3D OCT image, and Figure 5b,c contain cross-sectional images at different locations. In the results, no significant roughness could be observed on the tooth surface. Subsequently, the same tooth region was scanned with the UHR-OCT system. Figure 5d shows a 3D image obtained with UHR-OCT. Figure 5e,f depict the cross-sectional images at different locations. Moreover, Figure 5g,h show the *en-face* images at the depths indicated by the solid white and white-dashed lines in Figure 5f, and the cavity’s structures can be distinguished in Figure 5g,h. Moreover, surface roughness could be identified from Figure 5d. Thus, the cavity structures could be identified by the cross-sectional and depth-resolved *en-face* images. However, it was difficult to identify the cavity by using the conventional-resolution OCT systems with lower imaging resolutions.

### 3.2. Surface Roughness of Carious Teeth

The net demineralization reaction of dental caries can erode the appearance of the tooth’s surface, and the volumetric UHR-OCT image could observe the change in the roughness of the tooth’s surface. Figure 6 shows the 3D UHR-OCT images of the teeth’s surfaces. Figure 6a shows the smooth enamel surface of the normal tooth; Figure 6b,c show the representative carious surface images at the different early stages. The variation in their surface flatness can be clearly distinguished, as shown in Figure 6a–c. The smoothness of the enamel’s surface in normal and carious teeth was significantly different, showing the effect of demineralization on the tooth’s surface. In addition, as seen in Figure 6b,c, the UHR-OCT results further demonstrated that different early dental caries stages led to different degrees of erosion on the enamel’s surface. Figure 6d,e show a statistical analysis of the estimated surface roughness of normal and carious teeth in different early stages obtained by the conventional-resolution OCT and UHR-OCT, respectively. The number of teeth of each type collected in Figure 6d,e is at least 11 (*n* ≥ 11). The structural elements of the morphological filter in both systems were set to 70 μm to differentiate the demineralization-affected deformation from the general surface before calculating the surface roughness. Compared with the results of the conventional-resolution OCT system, the results of the UHR-OCT system, as shown in Figure 6e, show that the normal and carious teeth could be clearly differentiated, and the estimated roughness could also be used to distinguish the different early stages of caries.

### 3.3. Scattering Coefficient of Carious Teeth

As an optical imaging technology, OCT features the resolution of the longitudinal information of samples based on the scattering properties of tissue, enabling us to estimate the scattering coefficient of the sample. In the previous study, the OCT signal obtained from the demineralized teeth decreased significantly after passing through the surfaces, which caused an increase in their scattering coefficient compared with that of the normal teeth. Figure 7 shows the estimated scattering coefficients of teeth from conventional-resolution OCT and UHR-OCT. The results, presented in Figure 7a,b, showed good consistency with the previous reports mentioned above, proving that the scattering coefficients obtained with the OCT signal are an effective way to identify dental caries. However, the estimated scattering coefficients of the conventional-resolution OCT system, as shown in Figure 7a, had difficulty in distinguishing the different stages of early tooth decay. In contrast, Figure 7b shows the clear difference in the scattering coefficients estimated from the UHR-OCT signal between the first and second early stages of dental caries, showing the potential to quantify and analyze the evolution of early tooth decay with UHR-OCT.

### 3.4. Calculus Segmentation

In addition, to investigate the differences between caries and calculus, the teeth with calculus were scanned by both systems. Figure 8a,b represent the 3D and 2D images of a tooth with calculus obtained by the conventional-resolution OCT system, respectively. In contrast, Figure 8c–e show the 3D image and 2D images of the same tooth with calculus obtained by the UHR-OCT system. From Figure 8b, a thin film structure can be roughly identified, as indicated by the white arrow. In contrast, there is a clear and intact film structure on the tooth’s surface indicated by the white arrows in Figure 8d,e, corresponding to the calculus on the tooth’s surface. To estimate the thickness of the thin film’s structure, the boundaries of the upper and lower interfaces were obtained using the segmentation algorithm, and then the distribution of thickness of the calculus could be evaluated, as shown in Figure 8f. However, owing to the limited resolution of the conventional-resolution system, the thin film’s structure could not be differentiated easily, and estimating the thickness of calculus was difficult.

## 4. Discussion

Dental caries alters the tooth’s structure and causes changes in the tooth’s roughness. According to the results of Figure 6d,e, the average roughness of normal teeth is 1.7 μm; however, the average roughness increases to 3.7 and 6.6 μm in stages 1 and 2, respectively. This estimated roughness can be used to distinguish the different stages of caries. Furthermore, in previous reports, OCT has been used for estimating the scattering properties of dental caries and normal teeth [33,46]. However, due to the limited axial and transverse resolutions, identifying the tiny changes in the enamel’s layer in early-stage caries is still challenging, further limiting the accuracy of estimating the scattering coefficient. Thus, we proposed to use a UHR-OCT system with high axial and transverse resolutions of 2 and 1.8 μm, respectively, for the study of dental caries and calculus. Moreover, the results in Figure 7b illustrate that the average scattering coefficients of teeth with caries (*μ_s_* = 4.70 mm^−^^1^ and 5.65 mm^−^^1^) were significantly larger than that of the normal group. For the normal group, the average scattering coefficient of the UHR-OCT system’s results was 2.85 mm^−^^1^, which was greater than that of the conventional-resolution system (1.35 mm^−^^1^), probably resulting from the identification of much tinier structures with UHR-OCT. The estimated scattering coefficient of a tooth is related to the measured wavelength and materials (enamel or dentin), and the estimated result also varies due to different mathematical models and the stage of caries [47]. According to the proposed UHR-OCT system, the imaging depth of a tooth covers hundreds of micrometers. Therefore, the scattering coefficient of the enamel layer was estimated in this study. In a previous report, the measured scattering coefficient of enamel at 632 nm and 1050 nm was 6 and 1.5 mm^−^^1^, respectively [48]. Additionally, according to our UHR-OCT results, the average scattering coefficient of normal enamel is 2.85 mm^−^^1^, which is close to the result of the abovementioned report. Additionally, a conventional-resolution OCT system with a close central wavelength in comparison with the developed UHR-OCT system was implemented to reduce the additional effects caused by excessive differences in the wavelength. However, the slight difference in wavelength might be another factor responsible for the difference in the estimated scattering coefficients of the normal groups between the two OCT systems [49].

Both dental caries and calculus are produced by the interaction of liquid with bacteria and crystalline calcium phosphate in the oral cavity. They can be formed anywhere on the tooth’s surface; however, the formation process of these two is exactly the opposite [36]. The development of dental caries is a general demineralization reaction, but the formation of dental calculus is a mineralization process. These two lesions, generated in opposite ways from a mechanistic point of view, should theoretically be distinguishable from their internal structural patterns [6,50]. However, dental caries and calculus have a similar appearance and color in the early stages, and radiography is unable to distinguish between caries and calculus at this stage. Recently, research groups have used OCT to assess dental caries and dental calculus [33,51,52]. As the intensity of backscattered light of early dental caries and calculus is higher than that of sound tooth enamel tissue, it is effective for distinguishing between healthy and tooth enamel in the early stages of tooth disease. Although early dental caries and calculus are formed by exactly opposite processes, both are recognized as a “bright block” that cannot be differentiated by conventional-resolution OCT systems [53]. Therefore, distinguishing these two from standard OCT structural images is difficult. To observe the differences between dental caries and calculus, which are very similar upon visual inspection yet have an inverse relationship in the actual generation process, an improvement in the resolution was necessary.

In this study, UHR-OCT was proposed for imaging carious teeth in the early stages and to identify tiny morphological changes in the early stages, which are not easily identified by the conventional-resolution OCT systems. Since most of the past research has focused on apparent or larger areas of dental caries, an axial resolution of more than 10 microns was sufficient. With the proposed UHR-OCT system, the morphological structure of the demineralization or mineralization reaction could be clearly identified, which was not found in previous studies but is important for observing early lesions. While the pursuit of ultimate resolution is critical, the associated costs and challenges cannot be overlooked. Achieving high lateral resolution requires us to address the issues of distortions, compensations, and aberration, which inherently limit the field of view and imaging depth, thus impacting the observation of extensive or deep dental lesions [28]. Furthermore, high-magnification objectives typically come with short working distances or larger lens sizes, posing significant challenges to enhancing the system’s mobility.

In this study, we aimed to explore the simultaneous impact of high lateral and axial resolutions on the diagnosis of early dental lesions, a domain seemingly sparse in existing research. To fulfill this objective, we used a high numerical aperture (NA) objective, inherently constraining the system’s imaging range. Our findings indicated that the superior axial resolution of the UHR-OCT system aids in observing alternations in the internal tissue’s microstructure through cross-sectional imagery, while a high lateral resolution is crucial for examining *en-face* information such as surface roughness. Thus, there is merit in contemplating moderate adjustments to the NA of the objective lens to harmonize with the field of view (FOV).

Moreover, we recognized the substantial discrepancies between in vitro and in vivo conditions, particularly concerning the patient’s comfort, saliva flow, and artifacts of motion. We concede that extensive in vivo studies are imperative before this method can transition into clinical practice. As a result, developing a UHR-OCT system with a more extensive scanning range is a promising direction for clinical applications in early diagnosis and treatment. Previous reports have proposed various designs for handheld probes suitable for scanning the oral cavity [18,24,53,54,55]. For future clinical applications, the system could be modified into a catheter-based system or include a probe composed of the sample arm to enhance the maneuverability of the detection system. This adaptation would facilitate an efficient scanning process in diverse oral cavity scenarios.

## 5. Conclusions

Tooth caries and calculus are the most common problems in the oral cavity; however, identifying caries in the early stages and quantitatively evaluating its severity are challenging. In this study, we developed a UHR-OCT system for the study of caries and calculus. In comparison with the conventional OCT system with lower imaging resolutions, the structure of the cavity and the calculus layer in the early stages could be clearly identified with UHR-OCT. Moreover, quantitative analyses of tooth caries, including roughness and the scattering coefficient, were also proposed, and it can be noted that the roughness and scattering coefficient of the enamel layer increased when cavities were present. In addition, the thickness of calculus can be estimated from the results of UHR-OCT to evaluate the severity. The results indicate that UHR-OCT could be a potential tool for the early diagnosis of caries and calculus.

## Figures and Tables

**Figure 1 bioengineering-10-01317-f001:**
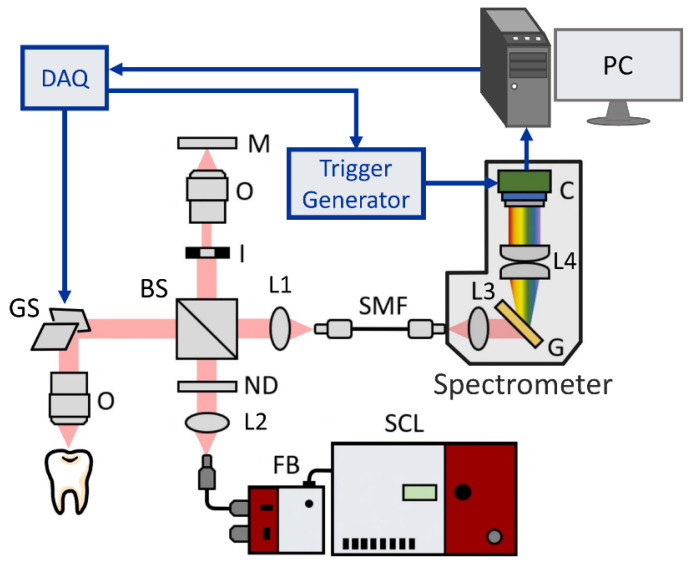
Schematic of our developed UHR-SD-OCT system. SCL, supercontinuum laser; FB, filter box; ND, neutral density filter; BS, beam splitter; I, iris diaphragm; GS, galvanometer; O, objective; L1–L4, lenses; M: mirror; SMF, single-mode fiber; G, transmission grating; C, line-scan camera; DAQ, data acquisition board.

**Figure 2 bioengineering-10-01317-f002:**
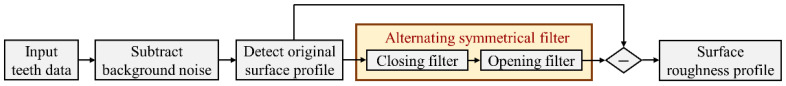
Flow chart of the process of acquiring the surface roughness profile.

**Figure 3 bioengineering-10-01317-f003:**
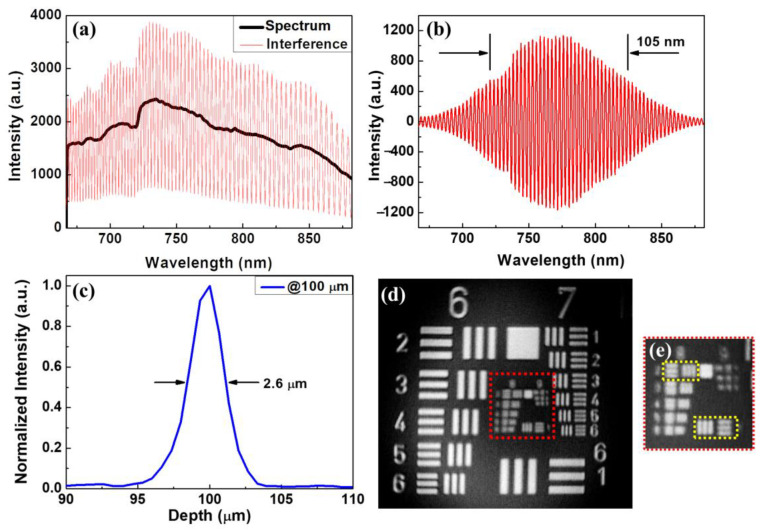
Interferogram acquired by the line-scan camera (**a**) before and (**b**) after the processes of DC removal, resampling, and window shaping. (**c**) Axial PSF measured in air and (**d**) OCT *en-face* image of the USAF resolution target. (**e**) Magnified image of the region indicated by the red square in (**d**). The upper left and lower right yellow squares in (**e**) are elements 1 and 2 in group 8 of the USAF resolution target.

**Figure 4 bioengineering-10-01317-f004:**
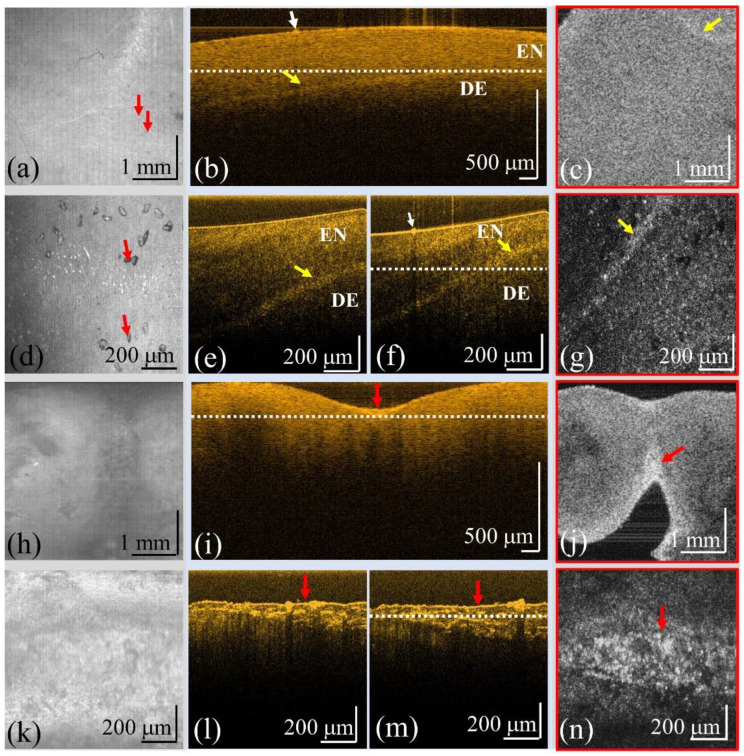
OCT results of a normal tooth obtained by (**a**–**c**) the conventional-resolution OCT system and (**d**–**g**) the UHR-OCT system, and OCT results of a tooth with caries obtained from (**h**–**j**) the conventional-resolution OCT system and (**k**–**n**) the UHR-OCT system. (**a**,**d**,**h**,**k**) The *en-face* images at the tooth’s surfaces, (**b**,**e**,**f**,**i**,**l**,**m**) the cross-sectional images, and (**c**,**g**,**j**,**n**) the *en-face* images at the depths indicated by the white lines in (**b**), (**f**), (**i**), and (**m**), respectively. The yellow and red arrows indicate the enamel–dentin boundary and cavities, respectively. The red arrows in (**a**,**d**) indicate the small damage on the tooth surface. EN, enamel; DE, dentin.

**Figure 5 bioengineering-10-01317-f005:**
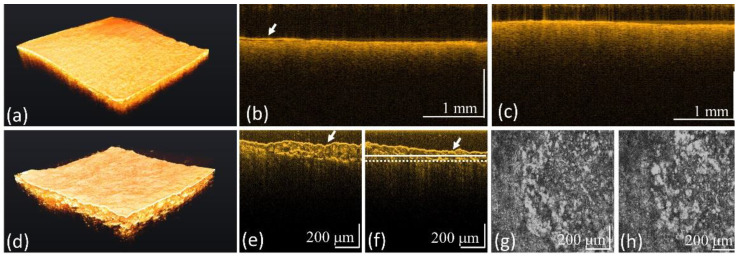
OCT results of the tooth with caries obtained from (**a**–**c**) the convectional-resolution OCT and (**d**–**h**) the UHR-OCT systems including (**a**,**d**) 3D, (**b**,**c**,**e**,**f**) 2D, and (**g**,**h**) *en-face* images at the depths indicated by the solid white and white-dashed lines in (**f**). The white arrows indicate the tooth’s surface. The 3D imaging areas of conventional-resolution OCT and UHR-OCT systems are 5 × 5 and 1 × 1 mm^2^, respectively.

**Figure 6 bioengineering-10-01317-f006:**
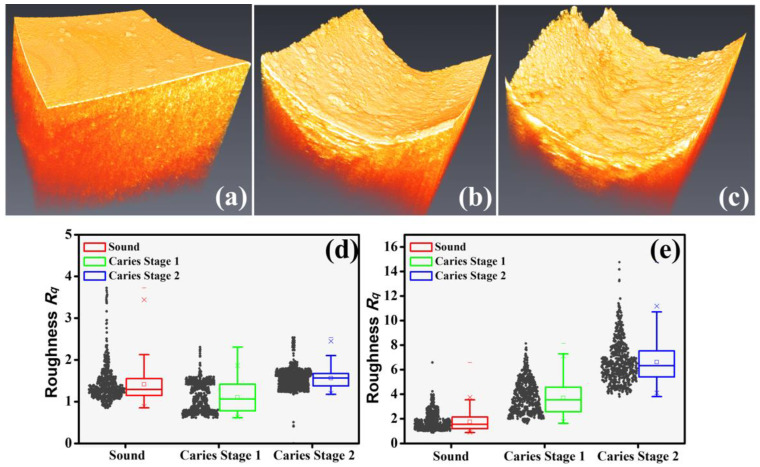
Three-dimensional UHR-OCT images of (**a**) the normal tooth; (**b**) early dental caries, stage 1; and (**c**) early dental caries, stage 2. Statistical results of the estimated surface roughness of carious teeth in the different early stages obtained by (**d**) the conventional-resolution OCT system and (**e**) the UHR-OCT system. The “×” symbols in each statistical result represent the upper and lower limits of the confidence interval at a 95% confidence level. The 3D imaging areas of the conventional-resolution OCT and UHR-OCT systems are 5 × 5 and 1 × 1 mm^2^, respectively.

**Figure 7 bioengineering-10-01317-f007:**
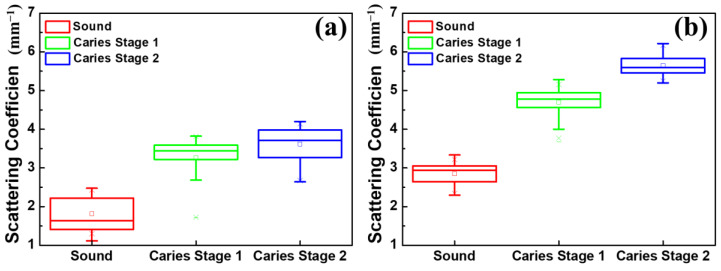
Scattering coefficients estimated from teeth at different stages using (**a**) the conventional-resolution OCT system and (**b**) the UHR-OCT system.

**Figure 8 bioengineering-10-01317-f008:**
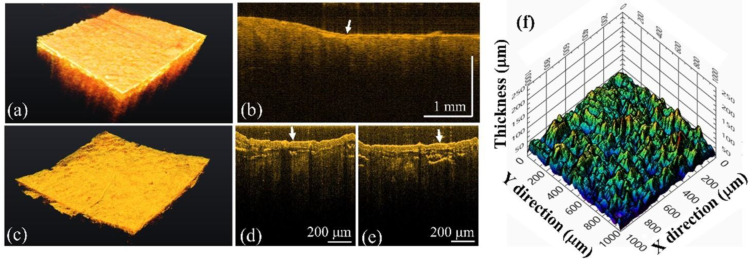
OCT results of the tooth with calculus tooth obtained using (**a**,**b**) the conventional-resolution OCT system and (**c**–**e**) the UHR-OCT systems including (**a**,**c**) 3D and (**b**,**d**,**e**) 2D images. (**f**) The distribution of thickness of the calculus. The white arrows indicate the tooth’s surface. The 3D imaging areas of the conventional-resolution OCT and UHR-OCT systems are 5 × 5 and 1 × 1 mm^2^, respectively.

**Table 1 bioengineering-10-01317-t001:** The specifications of the conventional-resolution OCT and the UHR-OCT used in this study.

	Center Wavelength	FWHM	Axial Resolution	Transverse Resolution	Image Area	Frame Rate
**Conventional-resolution OCT**	840 nm	45 nm	~7 μm	~10 μm	25 mm^2^	50 frames/s
**UHR-OCT**	775 nm	105 nm	~2.6 μm	~1.8 μm	1 mm^2^	50 frames/s

## Data Availability

Due to the regulation of IRB, data sharing is not applicable to this article.

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
