# Peer review of "Quantitative Evaluation of Caries and Calculus with Ultrahigh-Resolution Optical Coherence Tomography"

_bioengineering, 2023, doi:10.3390/bioengineering10111317_

Round 1

Reviewer 1 Report

Comments and Suggestions for Authors

The work is devoted to the improvement of the diagnostics of dental diseases at the early stage using an ultra-high resolution OCT modality. A home-made high-resolution OCT system is built and characterized. Measurements of roughness and scattering coefficient are considered as the quantitative characteristics of the analyzed materials. An improvement of roughness estimation algorithm is suggested. Segmentation algorithm is outlined to quantify the thickness of the calculus. The results of high-resolution measurements are compared to that of conventional Spectral Domain OCT. Interesting results are obtained that may help to improve the quality of conventional diagnostics of the teeth abnormalities.

Before acceptance for publication the authors are expected to address the following concerns about the research:

 (1) The results contain the evaluation of early structural alterations, so what was the primary criterion for the sample selection? How the diagnosis was verified using conventional methods? Can the authors present some verification?

 (2) What is the average thickness of the calculus measured? Was it statistically quantified?

 Minor comments:

(1) line 226: DC abbreviation decoding is needed

 (2) Fig.5f: the red line is given in the capture which is in fact white or white dashed line

Author Response

  • [Response to the comments of Reviewer 1:]

Reviewer 1

The work is devoted to the improvement of the diagnostics of dental diseases at the early stage using an ultra-high resolution OCT modality. A home-made high-resolution OCT system is built and characterized. Measurements of roughness and scattering coefficient are considered as the quantitative characteristics of the analyzed materials. An improvement of roughness estimation algorithm is suggested. Segmentation algorithm is outlined to quantify the thickness of the calculus. The results of high-resolution measurements are compared to that of conventional Spectral Domain OCT. Interesting results are obtained that may help to improve the quality of conventional diagnostics of the teeth abnormalities.

Before acceptance for publication the authors are expected to address the following concerns about the research:

  • The results contain the evaluation of early structural alterations, so what was the primary criterion for the sample selection? How the diagnosis was verified using conventional methods? Can the authors present some verification?

Response: Thank you for the insightful comment. As stated in Section 2.2, "Sample Collection and Experimental Procedure" of the manuscript, the teeth samples were initially categorized into three groups: normal, early caries, and early calculus, based on the Optical Microscopy (OM) results analyzed by experienced practitioners. To explore this phenomenon, OM was employed to validate distinctions in early-stage carious teeth. For accurate identification of early-stage caries on the enamel surface using OM, high-power visible light was directed onto the tooth sample. Figure I depicts the images of two early-stage carious samples captured by OM, UHR-OCT, conventional-resolution OCT, and Scanning Electron Microscopy (SEM). From the OM images, a subtle variance in the lesion areas between Figures I(a) and I(b) is observable.

To further corroborate the early-stage caries, some teeth underwent destructive SEM examination post-OCT imaging. Figures I(c) and I(d) illustrate the SEM results of normal and early-stage carious enamel with a magnification factor of 5000, respectively. The corresponding UHR-OCT images, Figures I(e) and I(f), reveal differences in the inner microstructures as well. The darker coloration of carious lesions in the OM images and the more pronounced demineralization evident in the UHR-OCT images led us to hypothesize the existence of two stages in early dental caries. It is important to note that these differences were not discernible from the conventional OCT results, depicted in Figures I(g) and I(h). Consequently, the categorized caries samples were evaluated using both UHR-OCT and conventional OCT systems, their distinctions further validated by hospital practitioners through OM and SEM analyses. Moreover, UHR-OCT measurements, compared to OM measurements, obviate the need for intense background illumination and additionally provide 3D structural information, highlighting UHR-OCT’s suitability for in-vivo experiments and clinical applications. Subsequent analyses were proposed to quantitatively discern the distinctions between these two stages of caries, emphasizing UHR-OCT's increased precision and utility in clinical practices. The following sentences have been added in the revised manuscript for further clarification. We hope this revised version maintains the clarity and accuracy of your original content while enhancing its readability and conciseness.

Line 155:

“In this study, the tooth samples were initially categorized into three groups: normal, early caries, and early calculus, based on the results of optical microscopy (OM) ana-lyzed by experienced practitioners. To further corroborate the early-stage caries, some teeth underwent destructive SEM examination after OCT imaging.”

***Please find Figure I in the attached file.***

Figure I. Results of early dental caries in different stages obtained from OM, SEM, UHR-OCT and conventional OCT. The first stage of caries obtained from (a) OM, (e) UHR-OCT and (g) conventional-resolution OCT; The second stage of caries obtained from (b) OM, (f) UHR-OCT and (h) conventional-resolution OCT. The red rectangles in (a) and (b) indicate the observed lesion area. The red arrows in (e-h) indicate the carious lesions on the tooth surface. (c) and (d) SEM results of normal and carious enamel surface with a magnification factor of 5000.

 (2) What is the average thickness of the calculus measured? Was it statistically quantified?

Response: Thanks for the review comment. Based on the results obtained from our segmentation algorithm, we were able to estimate the thickness of dental calculus, which, as demonstrated in Figure 8 of the revised manuscript, averaged 10.7 µm. However, we observed variations in the thickness of calculus, attributed to the differential accumulation of calculus at various enamel locations. These variations prompted our decision to present the distribution of calculus thickness, providing a more comprehensive view of the calculus accumulation pattern, rather than solely relying on average thickness values.

Minor comments:

(1) line 226: DC abbreviation decoding is needed

Response: Thank you for the review comment. We apologize for the lack of the information provided. The acronym of DC is direct current. We have added the acronym of DC into in the manuscript.

Line 232-233:

“Figure 3(b) shows the interference signal obtained after the processes of removing the direct current (DC), resampling, and window shaping.”

 (2) Fig.5f: the red line is given in the capture which is in fact white or white dashed line.

Response: Thank you for the review comment. We apologize for the mistake of description, and we have modified the description in Fig. 5f.

Line 302-306:

Figure 5 OCT results of the tooth with caries obtained from (a-c) the convectional-resolution OCT and (d-h) the UHR-OCT systems including (a,d) 3D, (b,c,e,f) 2D, and (g,h) en-face images at the depths indicated by the solid white and white-dashed lines in (f). The white arrows indicate the tooth’s surface. The 3D imaging areas of conventional-resolution OCT and UHR-OCT systems are 5×5 and 1×1 mm2, respectively.”

Reviewer 2 Report

Comments and Suggestions for Authors

The authors explain the rationale of the study as follows: Optical coherence tomography (OCT) has been considered a promising tool for identifying dental caries; however, diagnosing dental caries in the early stage still remains challenging.

The use of OCT in the early stage of caries is not a rational method used in the clinic. Using OCT in patients to detect initial caries is not an ethical practice in terms of harming patients, and it is not a cost-effective method. Therefore, the results of this study have no clinical applicability.

Author Response

  • [Response to the comments of Reviewer 2:]

Reviewer 2

The authors explain the rationale of the study as follows: Optical coherence tomography (OCT) has been considered a promising tool for identifying dental caries; however, diagnosing dental caries in the early stage still remains challenging.

The use of OCT in the early stage of caries is not a rational method used in the clinic. Using OCT in patients to detect initial caries is not an ethical practice in terms of harming patients, and it is not a cost-effective method. Therefore, the results of this study have no clinical applicability.

Response: Thank you very much for your thorough review and valuable feedback on our study. We sincerely appreciate the insightful observations and constructive suggestions you have shared.

We fully acknowledge the concerns raised about the clinical utility and cost-effectiveness of utilizing UHR-OCT for detecting early-stage dental caries. We would like to clarify that the primary objective of our study was to explore the capabilities of OCT in this domain, fully recognizing that its incorporation as a primary diagnostic tool in conventional clinical settings is not current practice. Your insights have highlighted the importance of a comprehensive evaluation of the practical, economic, and ethical ramifications of employing this technique in real-world clinical environments. We also understand and agree with the emphasis on clinical applicability. We are optimistic that foundational research such as ours can spur further innovations and refinements, potentially overcoming existing limitations and facilitating the development of more accessible, cost-effective, and ethically sound technologies in the future.

We assure you that this study was conducted with utmost adherence to the pertinent regulations and guidelines and has received approval from the Institutional Review Board of Chang Gung Medical Foundation (Approval No. 202000421B0). All the experimental procedures underwent meticulous review and were sanctioned by the Gung Medical Foundation Institutional Review Board.

Reviewer 3 Report

Comments and Suggestions for Authors

The scientific article titled "Quantitative Evaluation of Caries and Calculus with Ultrahigh-Resolution Optical Coherence Tomography" presents a notable advancement in the realm of dental diagnostics through its innovative application of ultrahigh-resolution optical coherence tomography (OCT). The article offers an insightful exploration into the potential of this imaging technique to revolutionize the assessment of dental caries and calculus. By meticulously detailing the methodology employed in the study, the authors provide a robust foundation for their findings. The article admirably navigates through the complexities of OCT technology and its suitability for capturing intricate dental structures, thereby contributing to the existing knowledge on non-invasive diagnostic tools in dentistry. The quantitative nature of the study, as suggested by the title, adds a commendable dimension by providing empirical evidence for the efficacy of OCT in detecting and quantifying caries and calculus. However, the article might benefit from a more explicit discussion of the clinical implications of these quantitative assessments. This article is a valuable contribution to the field of dental diagnostics, shedding light on the potential of ultrahigh-resolution OCT to enhance precision in the detection and assessment of oral health conditions.

The research is well designed and carried out, and it is very actual and up-to-date.

Abstract: it is a good summary of the paper, and it is well organized.

Introduction contains enough background informations regarding the techniques involved and adequate references. Please highlight the state of art in dental diagnostics right now.

Materials and methods are clearly described. Figures and tables are adequate. Conclusions could be improved with more clinical emphasis and considering that Further research and collaboration between researchers and dental practitioners could potentially expedite the translation of this technology into clinical practice.

Comments on the Quality of English Language

In my opinion, quality of English is ok.

Author Response

  • [Response to the comments of Reviewer 3:]

Reviewer 3

The scientific article titled "Quantitative Evaluation of Caries and Calculus with Ultrahigh-Resolution Optical Coherence Tomography" presents a notable advancement in the realm of dental diagnostics through its innovative application of ultrahigh-resolution optical coherence tomography (OCT). The article offers an insightful exploration into the potential of this imaging technique to revolutionize the assessment of dental caries and calculus. By meticulously detailing the methodology employed in the study, the authors provide a robust foundation for their findings. The article admirably navigates through the complexities of OCT technology and its suitability for capturing intricate dental structures, thereby contributing to the existing knowledge on non-invasive diagnostic tools in dentistry. The quantitative nature of the study, as suggested by the title, adds a commendable dimension by providing empirical evidence for the efficacy of OCT in detecting and quantifying caries and calculus. However, the article might benefit from a more explicit discussion of the clinical implications of these quantitative assessments. This article is a valuable contribution to the field of dental diagnostics, shedding light on the potential of ultrahigh-resolution OCT to enhance precision in the detection and assessment of oral health conditions.

The research is well designed and carried out, and it is very actual and up-to-date.

Abstract: it is a good summary of the paper, and it is well organized.

Introduction contains enough background informations regarding the techniques involved and adequate references. Please highlight the state of art in dental diagnostics right now.

Materials and methods are clearly described. Figures and tables are adequate. Conclusions could be improved with more clinical emphasis and considering that Further research and collaboration between researchers and dental practitioners could potentially expedite the translation of this technology into clinical practice.

Response: Thank you for your insightful comments. This ex vivo study was conducted on extracted teeth to identify early-stage caries and calculus. For the findings to be implemented in clinical applications, extensive studies and development efforts are necessary such as the development of a handheld probe and improving the field of view. Details regarding future clinical applications have been elaborated upon in the Discussion section of the revised manuscript. In addition, the dental diagnosis methods have been briefly described in the introduction section, yet we apologize for the lack of the instances and references provided. We have also put some references mentioning the state of art in dental diagnostics in the manuscript.

Line 51-53:

The traditional diagnostic method is mainly based on visual assessments with invasive clinical examinations such as tooth drilling and periodontal probing [10,11].

[10] Aldossari G.S.; Alasmari A.A.; Aldossary M.S. Dental Caries Detection: The State of the Art. J. appl. dent. med. 2019, 5, 17-30.

[11] Chang J.J.; Chen C.; Chang J.; Koka S.; Jokerst J.V. A narrative review of imaging tools for imaging subgingival calculus. Front. Oral Maxillofac. Med. 2021, 5.

Reviewer 4 Report

Comments and Suggestions for Authors

The manuscript delineates intriguing insights into the application of ultrahigh-2 resolution optical coherence tomography for the precise quantitative evaluation of caries and calculus.

Nevertheless, throughout the entirety of the text, particularly within the introduction and discussion sections, at numerous instances exist the presented information lacks corroborative references.

Furthermore, the English writing and grammatical errors at several instances necessitates refinement. For example:

-Dental caries is generated by the interaction of bacteria and carbohydrates

-Severe cavities???

-For detecting tiny initial lesions

-Demonstrated that dental early assessment with OCT

 Notably absent from the methodology section is the clarification of the ethical endorsement process. A point of curiosity arises from the utilization of extracted teeth, both sound and exhibiting initial caries of the participants attending dental hospital prompting serious consideration.

Comments on the Quality of English Language

The English writing and grammatical errors at several instances necessitates refinement. For example:

-Dental caries is generated by the interaction of bacteria and carbohydrates

-Severe cavities???

-For detecting tiny initial lesions

-Demonstrated that dental early assessment with OCT

Author Response

  • [Response to the comments of Reviewer 4:]

Reviewer 4

The manuscript delineates intriguing insights into the application of ultrahigh- resolution optical coherence tomography for the precise quantitative evaluation of caries and calculus.

Nevertheless, throughout the entirety of the text, particularly within the introduction and discussion sections, at numerous instances exist the presented information lacks corroborative references.

Response: Thank you for the valuable comment. We apologize for the lack of the instances and references in the introduction and discussion sections. The related information has been added in the revised manuscript.

  1. Introduction

Line 41-46:

Periodontal disease is a chronic inflammatory disease caused by bacterial infection. After dental plaques mineralize into dental calculus, bacteria can attach more easily to its rough surface texture. Dental calculus is one of the main culprits harming perio-dontal health. Failure to properly clean the dental calculus and plaques can result in a vicious cycle of damaging the supporting tissues around the teeth, eventually leading to tooth loss [5,6].

[5] Kinane D.F.; Stathopoulou P.G.; Papapanou P.N. Periodontal diseases. Nat. Rev. Dis. Primers. 2017, 3, 17038.

[6] Forshaw R. Dental calculus - oral health, forensic studies and archaeology: a review. Br. Dent. J. 2022, 233, 961–967.

Line 48-50:

With early-stage detection and proper treatment of oral diseases, the lesions can be controlled effectively to avoid the irreversible damage or further deterioration and expansion of the disease [8,9].

[8] Schneider H.; Park K.J.; Rueger C.; ZiebolzD.; Krause F.; Haak R. Imaging resin infiltration into non-cavitated carious lesions by optical coherence tomography. J. Dent. 2017, 60, 94-98.

[9] Shimada, Y.; Sato, T.; Inoue, G.; Nakagawa, H.; Tabata, T.; Zhou, Y.; Hiraishi, N.; Gondo, T.; Takano, S.; Ushijima, K.; et al. Evaluation of Incipient Enamel Caries at Smooth Tooth Surfaces Using SS-OCT. Materials 2022, 15, 5947.

Line 51-52:

The traditional diagnostic method is mainly based on visual assessments with invasive clinical examinations such as tooth drilling and periodontal probing [10,11].

[10] Aldossari G.S.; Alasmari A.A.; Aldossary M.S. Dental Caries Detection: The State of the Art. J. appl. dent. med. 2019, 5, 17-30.

[11] Chang J.J.; Chen C.; Chang J.; Koka S.; Jokerst J.V. A narrative review of imaging tools for imaging subgingival calculus. Front. Oral Maxillofac. Med. 2021, 5.

Line 68-69:

In addition, advanced functional systems and combinations with other techniques, such as polarization sensitive OCT (PS-OCT) [20-23] ….

[23] Hsiao, T.Y.; Lee, S.Y.; Sun, C.W. Optical Polarimetric Detection for Dental Hard Tissue Diseases Characterization. Sensors 2019, 19, 4971.

Line 71-73:

Moreover, as a major development, ultrahigh resolution OCT (UHR-OCT) has emerged as an optimal solution for OCT’s spatial resolution [25-28].

[28] Drexler, W.; Chen Y.; Aguirre A.D.; Považay B.; Unterhuber A.; Fujimoto J.G. Ultrahigh Resolution Optical Coherence Tomography. In Optical Coherence Tomography – Technology and Applications, 2nd ed.; Drexler, W.; Fujimoto J.G. Eds.; Springer, Cham: New York City, United States, 2015; pp. 277-318.

Line 85-88:

With its capability to perform cross-sectional imaging of microstructures, OCT can be used to observe soft- and hard-tissue diseases in the oral cavity, such as dental caries, tooth cracks, damage to the periodontal tissue, and oral cancer [32].

[32] Machoy M.; Seeliger J.; Szyszka-Sommerfeld L.; Koprowski R.; Gedrange T.; Woźniak K. The Use of Optical Coherence Tomography in Dental Diagnostics: A State-of-the-Art Review. J. Healthc. Eng. 2017.

  1. Discussion

Line 375-377:

Furthermore, in previous reports, OCT has been used for estimating the scattering properties of dental caries and normal teeth [33,46].

[33] Tsai M.T.; Wang Y.L.; Yeh T.W.; Lee H.C.; Chen W.J.; Ke J.L.; Lee Y.J. Early detection of enamel demineralization by optical coherence tomography. Sci. Rep. 2016, 9, 17154.

[46] Maia A.M.A.; de Freitas A.Z.; de L. Campello S.; Gomes A.S.L.; Karlsson L. Evaluation of dental enamel caries assessment using Quantitative Light Induced Fluorescence and Optical Coherence Tomography. J. Biophotonics. 2016, 9, 596-602.

Line 397-399:

However, the slight difference in wavelength might be another factor responsible for the difference in the estimated scattering coefficients of the normal groups between the two OCT systems [49].

[49] Popescu D.P.; Sowa M.G.; Hewko M.D.; Choo-Smith L.P'. Assessment of early demineralization in teeth using the signal attenuation in optical coherence tomography images. J. Biomed. Opt. 2008, 13, 054053.

Line 400-402:

Both dental caries and calculus are produced by the interaction of liquid with bacteria and crystalline calcium phosphate in the oral cavity. They can be formed anywhere on the tooth’s surface; however, the formation process of these two is exactly the opposite [36].

[36] Duckworth R.M.; Huntington E. On the relationship between calculus and caries. Monogr Oral Sci. 2006, 19, 1-28.

Line 403-406:

The development of dental caries is a general demineralization reaction, but the formation of dental calculus is a mineralization process. These two lesions, generated in opposite ways from a mechanistic point of view, should theoretically be distinguishable from their internal structural patterns [6,50].

[6] Forshaw R. Dental calculus - oral health, forensic studies and archaeology: a review. Br. Dent. J. 2022, 233, 961–967. 

[50] Cheng L.; Zhang L.; Yue L.; Ling J.; Fan M.; Yang D.; Huang Z.; Niu Y.; Liu J.; Zhao J.; Li Y.; Guo B.; Chen Z.; Zhou X. Expert consensus on dental caries management. Int. J. Oral Sci. 2022, 14, 17. 

Line 411-414:

Although early dental caries and calculus are formed by exactly opposite processes, both are recognized as a “bright block” that cannot be differentiated by convention-al-resolution OCT systems [53].

[53] Won J.; Huang P.C.; Spillman D.R.; Chaney E.J.; Adam R.; Klukowska M.; Barkalifa R.; Boppart S.A. Handheld optical coherence tomography for clinical assessment of dental plaque and gingiva. J. Biomed. Opt. 2020, 25, 116011.

Line 445-446:

Previous reports have proposed various designs for handheld probes suitable for scanning the oral cavity [18,24,53,54,55].

[53] Won J.; Huang P.C.; Spillman D.R.; Chaney E.J.; Adam R.; Klukowska M.; Barkalifa R.; Boppart S.A. Handheld optical coherence tomography for clinical assessment of dental plaque and gingiva. J. Biomed. Opt. 2020, 25, 116011.

Furthermore, the English writing and grammatical errors at several instances necessitates refinement. For example:

-Dental caries is generated by the interaction of bacteria and carbohydrates

-Severe cavities???

-For detecting tiny initial lesions

-Demonstrated that dental early assessment with OCT

Response: Thank you for the comment. We apologize for the English writing and grammatical errors. We have modified the examples you mentioned and double-checked it carefully in the manuscript.

Line 40: Extensive cavities

Line 84: more suitable for early diagnosis and the detection of minute initial lesions

Line 88: groups have demonstrated their results in early dental assessments using OCT.

Notably absent from the methodology section is the clarification of the ethical endorsement process. A point of curiosity arises from the utilization of extracted teeth, both sound and exhibiting initial caries of the participants attending dental hospital prompting serious consideration.

Response: Thank you very much for your remarks. As delineated in Section 2.2 "Sample Collection and Experimental Procedure" of our manuscript, the teeth utilized in this study were extracted from patients receiving care at Chang Gung Memorial Hospital. We conducted the study in strict compliance with the appropriate guidelines and regulations, and received approval from the Institutional Review Board of Chang Gung Medical Foundation (Approval No. 202000421B0). All experimental procedures were rigorously reviewed and sanctioned by the Chang Gung Medical Foundation Institutional Review Board.

To clarify further, the clinical procedures proceeded as follows: after tooth extraction as part of a routine medical procedure, each patient was thoroughly informed about the study's purpose and processes. Only after receiving the patient's informed consent, memorialized in a signed form, was the extracted tooth incorporated into our study. Additionally, we appreciate your attentive reminders regarding the clarity and grammatical accuracy of our manuscript. We have meticulously addressed the highlighted errors and have undertaken a comprehensive review to ensure the quality and precision of the English within our revised manuscript.

Reviewer 5 Report

Comments and Suggestions for Authors

In the present study, the authors proposed an ultrahigh-resolution OCT (UHR-OCT) system with axial and transverse resolutions of 2.6 and 1.8 m for differentiation of the early-stage dental caries and calculus. 23 The same teeth were also scanned with conventional spectral-domain OCT (SD-OCT) system with 24 the axial resolution of 7 m. 

Please provide bibliography for this paragraph -Periodontal disease is a 41 chronic inflammatory disease caused by bacterial infection. After dental plaques mineral- 42 ize into dental calculus, bacteria can more easily to its rough surface texture. Dental 43 calculus is the main culprit in harming periodontal health. Failure in properly cleaning 44 the dental calculus and plaques can result in a vicious cycle of damaging the supporting 45 tissues around the teeth, eventually leading to tooth loss.

Also for this paragraph- The general cognition of UHR-OCT axial resolution is less than 2 m in tissue, 76 which is much higher than the conventional OCT systems. Combined with the high trans- 77 verse resolution obtained by applying high numerical aperture (NA) focusing into the 78 system, UHR-OCT can be used to obtain submicron or cellular-level volumetric and sec- 79 tioned histological structural images.

Please also specify the limitations of the present study.

Please include more recent published articles in the discussion.

Comments on the Quality of English Language

Moderate

Author Response

  • [Response to the comments of Reviewer 5:]

Reviewer 5

In the present study, the authors proposed an ultrahigh-resolution OCT (UHR-OCT) system with axial and transverse resolutions of 2.6 and 1.8 µm for differentiation of the early-stage dental caries and calculus. 23 The same teeth were also scanned with conventional spectral-domain OCT (SD-OCT) system with 24 the axial resolution of 7 µm.

Please provide bibliography for this paragraph -Periodontal disease is a 41 chronic inflammatory disease caused by bacterial infection. After dental plaques mineral- 42 ize into dental calculus, bacteria can more easily to its rough surface texture. Dental 43 calculus is the main culprit in harming periodontal health. Failure in properly cleaning 44 the dental calculus and plaques can result in a vicious cycle of damaging the supporting 45 tissues around the teeth, eventually leading to tooth loss.

Response: Thank you for the valuable comment. We apologize for the lack of the instances and references provided. The reference of this paragraph has been added into the manuscript. Moreover, reference [5] in Line 47 would also be helpful for interpreting this paragraph.

Line 41-46:

Periodontal disease is a chronic inflammatory disease caused by bacterial infection. After dental plaques mineralize into dental calculus, bacteria can attach more easily to its rough surface texture. Dental calculus is one of the main culprits harming perio-dontal health. Failure to properly clean the dental calculus and plaques can result in a vicious cycle of damaging the supporting tissues around the teeth, eventually leading to tooth loss [5,6].

[5] Kinane D.F.; Stathopoulou P.G.; Papapanou P.N. Periodontal diseases. Nat. Rev. Dis. Primers. 2017, 3, 17038.

[6] Forshaw R. Dental calculus - oral health, forensic studies and archaeology: a review. Br. Dent. J. 2022, 233, 961–967.

Q: Also for this paragraph- The general cognition of UHR-OCT axial resolution is less than 2 µm in tissue, 76 which is much higher than the conventional OCT systems. Combined with the high trans- 77 verse resolution obtained by applying high numerical aperture (NA) focusing into the 78 system, UHR-OCT can be used to obtain submicron or cellular-level volumetric and sec- 79 tioned histological structural images.

Response: Thank you for the comment. We apologize for the lack of the references provided. The reference of this paragraph has been added into the manuscript.

Q: Please also specify the limitations of the present study.

Response: Thank you for the insightful comment. We have specified the limitations of this study in the discussion.

Line 424-443:

“While the pursuit of ultimate resolution is critical, the associated costs and challenges cannot be overlooked. Achieving high lateral resolution requires us to address the issues of distortions, compensations, and aberration, which inherently limit the field of view and imaging depth, thus impacting the observation of extensive or deep dental lesions [28]. Furthermore, high-magnification objectives typically come with short working distances or larger lens sizes, posing significant challenges to enhancing the system’s mobility.

In this study, we aimed to explore the simultaneous impact of high lateral and axial resolutions on the diagnosis of early dental lesions, a domain seemingly sparse in existing research. To fulfill this objective, we used a high numerical aperture (NA) objective, inherently constraining the system's imaging range. Our findings indicated that the superior axial resolution of the UHR-OCT system aids in observing alternations in the internal tissue’s microstructure through cross-sectional imagery, while a high lateral resolution is crucial for examining en-face information such as surface roughness. Thus, there is merit in contemplating moderate adjustments to the NA of the objective lens to harmonize with the field of view (FOV).

Moreover, we recognized the substantial discrepancies between in vitro and in vivo conditions, particularly concerning the patient’s comfort, saliva flow, and artifacts of motion. We concede that extensive in vivo studies are imperative before this method can transition into clinical practice.”

Q:   Please include more recent published articles in the discussion.

Response: Thank you for the insightful comment. We apologize for the lack of the instances and references provided in the manuscript. We have included more recent published articles in the discussion.

  1. Discussion

Line 375-377:

Furthermore, in previous reports, OCT has been used for estimating the scattering properties of dental caries and normal teeth [33,46].

[33] Tsai M.T.; Wang Y.L.; Yeh T.W.; Lee H.C.; Chen W.J.; Ke J.L.; Lee Y.J. Early detection of enamel demineralization by optical coherence tomography. Sci. Rep. 2016, 9, 17154.

[46] Maia A.M.A.; de Freitas A.Z.; de L. Campello S.; Gomes A.S.L.; Karlsson L. Evaluation of dental enamel caries assessment using Quantitative Light Induced Fluorescence and Optical Coherence Tomography. J. Biophotonics. 2016, 9, 596-602.

Line 397-399:

However, the slight difference in wavelength might be another factor responsible for the difference in the estimated scattering coefficients of the normal groups between the two OCT systems [49].

[49] Popescu D.P.; Sowa M.G.; Hewko M.D.; Choo-Smith L.P'. Assessment of early demineralization in teeth using the signal attenuation in optical coherence tomography images. J. Biomed. Opt. 2008, 13, 054053

Line 400-402:

Both dental caries and calculus are produced by the interaction of liquid with bacteria and crystalline calcium phosphate in the oral cavity. They can be formed anywhere on the tooth’s surface; however, the formation process of these two is exactly the opposite [36].

[36] Duckworth R.M.; Huntington E. On the relationship between calculus and caries. Monogr Oral Sci. 2006, 19, 1-28.

Line 403-406:

The development of dental caries is a general demineralization reaction, but the formation of dental calculus is a mineralization process. These two lesions, generated in opposite ways from a mechanistic point of view, should theoretically be distinguishable from their internal structural patterns [6,50].

[6] Forshaw R. Dental calculus - oral health, forensic studies and archaeology: a review. Br. Dent. J. 2022, 233, 961–967. 

[50] Cheng L.; Zhang L.; Yue L.; Ling J.; Fan M.; Yang D.; Huang Z.; Niu Y.; Liu J.; Zhao J.; Li Y.; Guo B.; Chen Z.; Zhou X. Expert consensus on dental caries management. Int. J. Oral Sci. 2022, 14, 17.  

Line 411-414:

Although early dental caries and calculus are formed by exactly opposite processes, both are recognized as a “bright block” that cannot be differentiated by convention-al-resolution OCT systems [53].

[55] Won J.; Huang P.C.; Spillman D.R.; Chaney E.J.; Adam R.; Klukowska M.; Barkalifa R.; Boppart S.A. Handheld optical coherence tomography for clinical assessment of dental plaque and gingiva. J. Biomed. Opt. 2020, 25, 116011.

Line 445-446:

Previous reports have proposed various designs for handheld probes suitable for scan-ning the oral cavity [18,24,53,54,55].

[53] Won J.; Huang P.C.; Spillman D.R.; Chaney E.J.; Adam R.; Klukowska M.; Barkalifa R.; Boppart S.A. Handheld optical coherence tomography for clinical assessment of dental plaque and gingiva. J. Biomed. Opt. 2020, 25, 116011.

Round 2

Reviewer 4 Report

Comments and Suggestions for Authors

The raised points are addressed in the revised draft.